# Nearly Tight Bounds For
# Differentially Private Multiway Cut

**Mina Dalirrooyfard**[*]
Machine Learning Research
Morgan Stanley
mina.dalirrooyfard@morganstanley.com

**Slobodan Mitrovic**[*]
Department of Computer Science
UC Davis
smitrovic@ucdavis.edu

**Yuriy Nevmyvaka**
Machine Learning Research
Morgan Stanley
yuriy.nevmyvaka@morganstanley.com

## Abstract

Finding min $s$-$t$ cuts in graphs is a basic algorithmic tool, with applications in image segmentation, community detection, reinforcement learning, and data clustering. In this problem, we are given two nodes as terminals and the goal is to remove the smallest number of edges from the graph so that these two terminals are disconnected. We study the complexity of differential privacy for the min $s$-$t$ cut problem and show nearly tight lower and upper bounds where we achieve privacy at no cost for running time efficiency. We also develop a differentially private algorithm for the multiway $k$-cut problem, in which we are given $k$ nodes as terminals that we would like to disconnect. As a function of $k$, we obtain privacy guarantees that are exponentially more efficient than applying the advanced composition theorem to known algorithms for multiway $k$-cut. Finally, we empirically evaluate the approximation of our differentially private min $s$-$t$ cut algorithm and show that it almost matches the quality of the output of non-private ones.

## 1 Introduction

Min $s$-$t$ cut, or more generally Multiway $k$-cut, is a fundamental problem in graph theory and occupies a central place in combinatorial optimization. Given a weighted graph and $k$ terminals, the multiway $k$-cut problem asks to divide the nodes of the graph into $k$ partitions such that (1) each partition has exactly one terminal, and (2) the sum of the weights of edges between partitions, known as the *cut value*, is minimized [9]. Multiway $k$-cut is a clustering method used in a large variety of applications, including energy minimization [21] and image segmentation [27, 32] in vision, reinforcement learning [25], community detection [33, 29] and many other learning and clustering tasks [4, 5, 30, 23].

The above applications are carried on large data sets, and more and more frequently those applications are executed on sensitive data. Hence, designing algorithms that preserve the privacy of data has been given substantial attention recently. A widely used and conservative standard of data privacy is *differential privacy (DP)* developed by Dwork [12] which indicates that an algorithm is differentially private if, given two *neighboring* databases, it produces statistically indistinguishable outputs. When DP is applied to graph data, two variants of DP were introduced [19]: in edge DP, two neighboring graphs differ in only one edge, and in node DP, two graphs are neighboring if one graph is obtained

---

[*]These authors contributed equally to this work

37th Conference on Neural Information Processing Systems (NeurIPS 2023).

from removing one node and all the edges incident to that node from the other graph. In this work, we focus on the edge DP. In a large fraction of cases, the nodes of a network are public but the edge attributes – which are the relationships between nodes – are private.

For many clustering algorithms such as $k$-means, $k$-median, and correlation clustering, tight or near-tight differentially private algorithms have already been developed [31, 17, 20, 8]. In particular, tight algorithms for differentially private *min-cut* has been long known [18], where the min-cut problem asks to divide the graph into two partitions such that the cut value is minimized; in the min-cut problem, there is no restriction that two nodes must be on different sides of the cut. Even though min-cut and min $s$-$t$ cut might seem similar, the algorithm techniques for min-cut do not extend to min $s$-$t$ cut. A glimpse of this difference can be seen in the fact that there exist polynomially many min-cuts in any graph, but there can be exponentially many min $s$-$t$ cuts for fixed terminals $s$ and $t$[*].

**Our Results and Technical Overview**    In this paper, we provide the edge-differentially private algorithm for min $s$-$t$ cut and prove that it is almost tight. To the best of our knowledge, this is the first DP algorithm for the min $s$-$t$ cut problem.

Our first result is an $\epsilon$-private algorithm for min $s$-$t$ cut with $O(\frac{n}{\epsilon})$ additive error.

**Theorem 1.1.** *For any $\epsilon > 0$ and for weighted undirected graphs, there is an algorithm for min $s$-$t$ cut that is $(\epsilon, 0)$-private and with high probability returns a solution with additive error $O(\frac{n}{\epsilon})$. Moreover, the running time of this private algorithm is the same as the running time of the fastest non-private one.*

Moreover, our proof of Theorem 1.1 extends to when an edge-weight changes by at most $\tau$ in two neighboring graphs. In that case, our algorithm is $(\epsilon, 0)$-private with $O(n \cdot \tau/\epsilon)$ additive error.

Furthermore, our approach uses the existing min $s$-$t$ cut algorithms in a black-box way. More precisely, our method changes the input graph in a specific way, i.e., it adds $2(n - 1)$ edges with special weight, and the rest of the processing is done by an existing algorithm. In other words, our approach automatically transfers to any computation setting – centralized, distributed, parallel, streaming – as long as there is a non-private min $s$-$t$ cut for that specific setting. The main challenge with this approach is showing that it actually yields privacy. Perhaps surprisingly, our approach is almost optimal in terms of the approximation it achieves. Specifically, we complement our upper-bound by the following lower-bound result.

**Theorem 1.2.** *Any $(\epsilon, \delta)$-differential private algorithm for min $s$-$t$ cut on $n$-node graphs requires expected additive error of at least $n/20$ for any $\epsilon \leq 1$ and $\delta \leq 0.1$.*

Note that Theorem 1.2 holds regardless of the graph being weighted or unweighted. Given an unweighted graph $G$, let the cut $C_s(G)$ be the $s$-$t$ cut where node $s$ is in one partition and all the other nodes are in the other partition. The algorithm that always outputs $C_s(G)$ is private, and has the additive error of $O(n)$ since the value of $C_s(G)$ is at most $n - 1$ in unweighted graphs. Thus one might conclude that we do not need Theorem 1.1 for a private algorithm, as one cannot hope to do better than $O(n)$ additive error by Theorem 1.2. However, for weighted graphs, this argument fails when $C_s(G)$ has a very large value due to weighted edge incident to $s$, or when a graph has parallel unweighted edges. In Section 5, we show an application in which one computes a weighted min $s$-$t$ cut while as the input receiving an unweighted graph. We further evaluate our theoretical results on private min $s$-$t$ cut in Section 5 and show that despite our additive approximation, our approach outputs a cut with the value fairly close to the min $s$-$t$ cut. Moreover, we show that it is in fact more accurate than more natural heuristics (such as outputting $C_s(G)$) for a wide range of the privacy parameter $\epsilon$.

Theorem 1.1 and Theorem 1.2 depict an interesting comparison to differentially private min cut algorithms. Gupta et al [18] provide an $\epsilon$-private algorithm with $O(\log n/\epsilon)$ additive error for min cut and shows that any $\epsilon$-private algorithm requires $\Omega(\log n)$ additive error. The large gap between

---

[*]Take $n - 2$ nodes $v_1, ..., v_{n-2}$ in addition to $s$ and $t$, and add $v_i$ to both $s$ and $t$ for all $i$. Suppose the weight of all edges is 1. There are $2^{n-2}$ min $s$-$t$ cuts, as each min s-t cut should remove exactly one edge from each $s - v_i - t$ path; either $(s, v_i)$ or $(v_i, t)$ is fine. In this example, there are $n - 2$ min cuts: for each $i$, remove edges $(v_i, s)$ and $(v_i, t)$ and this would be a cut of size 2. Moreover, one can show that the number of min-cuts is polynomial for any graph, there are at most $\binom{n}{2}$; please see [11].

$\epsilon$-private algorithms for the min cut and min $s$-$t$ cut could provide another reason one cannot easily extend algorithms for one to the other problem.

While finding (non-private) Min $s$-$t$ cut is polynomially solvable using max-flow algorithms [1, 7], the (non-private) Multiway $k$-cut problem is NP-hard for $k \geq 3$ [10] and finding the best approximation algorithm for it has been an active area of research [10, 6, 2, 24]. The best approximation factor for the Multiway cut problem is $1.2965$ [28] and the largest lower bound is $1.20016$ [3, 24]. From a more practical point of view, there are a few simple 2-approximation algorithm for multiway $k$-cut such as greedy splitting [34] which splits one partition into two partitions in a sequence of $k-1$ steps. Another simple 2-approximation algorithm by [10] is the following: (1) if the terminals of the graph are $s_1, \ldots, s_k$, for each $i$, contract all the terminals except $s_i$ into one node, $t_i$; (2) then run min $s$-$t$ cut on this graph with terminals $s_i, t_i$; (3) finally output the union of all these $k$ cuts. This algorithm reduces multiway $k$-cut into $k$ instances of min $s$-$t$ cut. Hence it is easy to see that if we use a $\epsilon$-private min $s$-$t$ cut with additive error $r$, we get a $k\epsilon$-private multiway $k$-cut algorithm with additive error $kr$ and multiplicative error 2. In a similar way one can make the greedy splitting algorithm private with same error guarantees. Using *advanced composition* [15, 16] can further reduce this dependency polynomially at the expense of private parameters. We design an algorithm that reduces the dependency on $k$ of the private multiway cut algorithm to $\log k$, which is exponentially smaller dependence than applying the advanced composition theorem. We obtain this result by developing a novel 2-approximation algorithm for multiway $k$-cut that reduces this problem to $\log k$ instances of min $s$-$t$ cut. Our result is represented in Theorem 1.3 and we defer the proof to supplementary materials. We note that the running time of the Algorithm of Theorem 1.3 is at most $O(\log k)$ times the private min $s$-$t$ cut algorithm of Theorem 1.1.

**Theorem 1.3.** *For any $\epsilon > 0$, there exists an $(\epsilon, 0)$-private algorithm for multiway $k$-cut on weighted undirected graphs that with probability $1 - 1/n$ at least returns a solution with value $2 \cdot OPT + O(n \log k / \epsilon)$, where $OPT$ is the value of the optimal non-private multiway $k$-cut.*

Finally, we emphasize that all our private algorithms output cuts as opposed to only their value. For outputting a number $x$ privately, the standard approach is to add Laplace noise $\mathrm{Lap}(1/\epsilon)$ to get an $\epsilon$-private algorithm with additive error at most $O(1/\epsilon)$ with high probability.

**Additional implications of our results.** First, the algorithm for Theorem 1.1, i.e., Algorithm 1, uses a non-private min $s$-$t$ cut algorithm in a black-box manner. Hence, Algorithm 1 essentially translates to any computation setting without asymptotically increasing complexities compared to non-private algorithms. Second, Algorithm 2 creates (recursive) graph partitions such that each vertex and edge appears in $O(\log k)$ partitions. To see how it differs from other methods, observe that the standard splitting algorithm performs $k-1$ splits, where some vertices appear in each of the split computations. There are also LP-based algorithms for the multiway $k$-cut problem, but to the best of our knowledge, they are computationally more demanding than the 2-approximate greedy splitting approach. As a consequence, in the centralized and parallel settings such as PRAM, in terms of total work, Algorithm 2 depends only logarithmically on $k$ while the popular greedy splitting algorithm has a linear dependence on $k$. To the best of our knowledge, no existing method matches these guarantees of Algorithm 2.

## 2 Preliminaries

In this section, we provide definitions used in the paper. We say that an algorithm $\mathcal{A}$ outputs an $(\alpha, \beta)$-approximation to a minimization problem $P$ for $\alpha \geq 1, \beta \geq 0$, if on any input instance $I$, we have $OPT(I) \leq \alpha \mathcal{A}(I) + \beta$ where $\mathcal{A}(I)$ is the output of $\mathcal{A}$ on input $I$ and $OPT(I)$ is the solution to problem $P$ on input $I$. We refer to $\alpha$ and $\beta$ as the multiplicative and additive errors respectively.

**Graph Cuts** We use $uv$ to refer to the edge between nodes $u$ and $v$. Let $G = (V, E, w_G)$ be a weighted graph with node set $V$, edge set $E$ and $w_G : V^2 \rightarrow \mathbb{R}_{\geq 0}$, where for all edges $uv \in E$, $w_G(uv) > 0$, and for all $uv \notin E$, $w_G(uv) = 0$. For any set of edges $C$, let $w_G(C) = \sum_{e \in C} w_G(e)$. If removing $C$ disconnects the graph, we refer to $C$ as a cut set. When $C$ is a cut set, we refer to $w_G(C)$ as the weight or value of $C$. We drop the subscript $G$ when it is clear from the context. For subsets $A, B \subseteq V$, let $E(A, B)$ be the set of edges $uv = e \in E$ such that $u \in A$ and $v \in B$.

**Definition 2.1** (Multiway $k$-cut). *Given $k$ terminals $s_1, \ldots, s_k$, a multiway $k$-cut is a set of edges $C \subseteq E$ such that removing the edges in $C$ disconnects all the terminals. More formally, there*

exists $k$-disjoint node subsets $V_1, \dots, V_k \in V$ such that $s_i \in V_i$ for all $i$, $\cup_{i=1}^k V_i = V$, and $C = \cup_{i,j \in \{1,\dots,k\}} E(V_i, V_j)$. *The* Multiway $k$-cut problem *asks for a Multiway $k$-cut with the lowest value.*

In our algorithms, we use the notion of *node contractions* which we formally define here.

**Definition 2.2** (Node contractions). *Let $Z \subseteq V$ be a subset of nodes of the graph $G = (V, E, w_G)$. By contracting the nodes in $Z$ into node $t^\star$, we add a node $t^\star$ to the graph and remove all the nodes in $Z$ from the graph. For every $v \in V \setminus Z$, the weight of the edge $t^\star v$ is equal to $\sum_{z \in Z} w_G(vz)$. Note that if none of the nodes in $Z$ has an edge to $v$, then there is no edge from $t^\star$ to $v$.*

**Differential Privacy** We formally define neighboring graphs and differential private algorithms.

**Definition 2.3** (Neighboring Graphs). *Graphs $G = (V, E, w)$ and $G' = (V, E', w')$ are called neighboring if there is $uv \in V^2$ such that $|w_G(uv) - w_{G'}(uv)| \leq 1$ and for all $u'v' \neq uv$, $u'v' \in V^2$, we have $w_G(u'v') = w_{G'}(u'v')$.*

**Definition 2.4** (Differential Privacy[12]). *A (randomized) algorithm $\mathcal{A}$ is $(\epsilon, \delta)$-private (or $(\epsilon, \delta)$-DP) if for any neighboring graphs $G$ and $G'$ and any set of outcomes $O \subset Range(\mathcal{A})$ it holds*

$$\Pr\left[\mathcal{A}(G) \in O\right] \leq e^\epsilon \Pr\left[\mathcal{A}(G') \in O\right] + \delta.$$

*When $\delta = 0$, algorithm $\mathcal{A}$ is* pure differentially private*, or $\epsilon$-private.*

**Definition 2.5** (Basic composition [14, 13]). *Let $\epsilon_1, \dots, \epsilon_t > 0$ and $\delta_1, \dots, \delta_t \geq 0$. If we run $t$ (possibly adaptive) algorithms where the $i$-th algorithm is $(\epsilon_i, \delta_i)$-DP, then the entire algorithm is $(\epsilon_1 + \dots + \epsilon_t, \delta_1 + \dots + \delta_t)$-DP.*

**Exponential distribution.** For $\lambda > 0$, we use $X \sim \text{Exp}(\lambda)$ to denote that a random variable $X$ is drawn from the exponential distribution with parameter $\lambda$. In our proofs, we use the following fact.

**Fact 2.1.** *Let $X \sim \text{Exp}(\lambda)$ and $z \geq 0$. Then $\Pr\left[X \geq z\right] = \exp\left(-\lambda z\right).$*

**Laplace distribution.** We use $X \sim \text{Lap}(b)$ to denote that $X$ is a random variable is sampled from the Laplace distribution with parameter $b$.

**Fact 2.2.** *Let $X \sim \text{Lap}(b)$ and $z > 0$. Then*

$$\Pr\left[X > z\right] = \frac{1}{2} \exp\left(-\frac{z}{b}\right) \ \text{and} \ \Pr\left[|X| > z\right] = \exp\left(-\frac{z}{b}\right).$$

**Fact 2.3.** *Let $X, Y \sim \text{Exp}(\epsilon)$. Then, the distribution $X - Y$ follows $\text{Lap}(1/\epsilon)$.*

We use $f_{\text{Lap}}^b$ to denote the cumulative distribution function of $\text{Lap}(b)$, i.e., $f_{\text{Lap}}^b(x) = \frac{1}{2b} \exp\left(-\frac{|x|}{b}\right)$.
We use $F_{\text{Lap}}^b$ to denote the cumulative distribution function of $\text{Lap}(b)$, i.e.,

$$F_{\text{Lap}}^b(x) = \begin{cases} \frac{1}{2} \exp\left(\frac{x}{b}\right) & \text{if } x \leq 0 \\ 1 - \frac{1}{2} \exp\left(-\frac{x}{b}\right) & \text{otherwise} \end{cases}$$

When it is clear from the context what $b$ is, we will drop it from the superscript.

**Claim 1.** *For any $t \in \mathbb{R}$ and $\tau \geq 0$ it holds*

$$\frac{F_{\text{Lap}}^{1/\epsilon}(t + \tau)}{F_{\text{Lap}}^{1/\epsilon}(t)} \leq e^{\tau \epsilon} \ \text{and} \ \frac{f_{\text{Lap}}^{1/\epsilon}(t + \tau)}{f_{\text{Lap}}^{1/\epsilon}(t)} \leq e^{\tau \epsilon}.$$

The proof of this claim is standard and, for completeness, appears in the Supplementary material.

## 3 DP Algorithm for Min $s$-$t$ Cut

In this section, we prove Theorem 1.1. Our DP min $s$-$t$ cut algorithm is quite simple and is provided as Algorithm 1. The approach simply adds an edge between $s$ and every other node, and an edge between $t$ and every other node. These edges have weight drawn from $\text{Exp}(\epsilon)$. The challenging part is showing that this simple algorithm actually preserves privacy. Moreover, when we couple the

---

**Algorithm 1:** Given a weighted graph $G$, this algorithm outputs a DP min $s$-$t$ cut.

---

**Input** : A weighted graph $G = (V, E, w)$
Terminals $s$ and $t$
Privacy Parameter $\epsilon$

**1 for** $u \in V \setminus \{s, t\}$ **do**

**2** $\quad$ Sample $X_{s,u} \sim \mathrm{Exp}(\epsilon)$ and $X_{t,u} \sim \mathrm{Exp}(\epsilon)$.

**3** $\quad$ In $G$ add an edge between $s$ and $u$ with weight $X_{s,u}$, and an edge between $t$ and $u$ with
$\quad$ weight $X_{t,u}$.

**4 return** the min $s$-$t$ cut in (the modified) $G$

---

approximation guarantee of Algorithm 1 with our lower-bound shown in the Supplementary material, then up to a $1/\epsilon$ factor Algorithm 1 yields optimal approximation guarantee.

*Remark:* Technically, there might be multiple min $s$-$t$ cuts that can be returned on Line 4 of Algorithm 1. We discuss that, without loss of generality, it can be assumed that there is a unique one. We elaborate on this in the Supplementary material.

### 3.1 Differential Privacy Analysis

Before we dive into DP guarantees of Algorithm 1, we state the following claim that we use in our analysis. Its proof is given in the Supplementary material.

**Lemma 3.1.** *Suppose $x$ and $y$ are two independent random variables drawn from $\mathrm{Lap}(1/\epsilon)$. Let $\alpha, \beta, \gamma$ be three fixed real numbers, and let $\tau \geq 0$. Define*

$$P(\alpha, \beta, \gamma) \stackrel{\mathrm{def}}{=} \Pr\left[x < \alpha, y < \beta, x + y < \gamma\right].$$

*Then, it holds that*

$$1 \leq \frac{P(\alpha + \tau, \beta + \tau, \gamma)}{P(\alpha, \beta, \gamma)} \leq e^{4\tau\epsilon}.$$

**Lemma 3.2.** *Let $\tau$ be the edge-weight difference between two neighboring graphs. Algorithm 1 is $(4\tau\epsilon, 0)$-DP.*

*Proof.* Let $G$ and $G'$ be two neighboring graphs. Let edge $e = uv$ be the one for which $w_e$ in $G$ and $G'$ differ; recall that it differs by at most $\tau$. To analyze the probability with which Algorithm 1 outputs the same cut for input $G$ as it outputs for input $G'$, we first sample all the $X_{s,x}$ and $X_{t,x}$ for $x \in V \setminus \{s, t, u, v\}$. Intuitively, the outcomes of those random variables are not crucial for the difference between outputs of Algorithm 1 invoked on $G$ and $G'$. We now elaborate on that.

Consider **all** the $s$-$t$ cuts – not only the minimum ones – in $G$ before $X_{s,u}$, $X_{s,v}$, $X_{t,u}$ and $X_{t,v}$ are sampled; for instance, assume for a moment that those four random variables equal $0$. Let $\mathcal{C}_{\mathrm{all-st-cuts}}(G)$ be all those cuts sorted in a non-decreasing order with respect to their weight. Observe that $\mathcal{C}_{\mathrm{all-st-cuts}}(G) = 2^{n-2}$. As we will show next, although there are exponentially many cuts, we will point to only four of them as crucial for our analysis. This will come in very handy in the rest of this proof. We now partition $\mathcal{C}_{\mathrm{all-st-cuts}}(G)$ into four groups based on which $u$ and $v$ are on the same side of the cut as $s$. Let $\mathcal{C}_u(G)$ be the subset of $\mathcal{C}_{\mathrm{all-st-cuts}}(G)$ for which $u$ is on the same side of a cut as $s$ while $v$ is on the same side of the cut as $t$. Analogously we define $\mathcal{C}_v(G)$. We use $\mathcal{C}_{u,v}(G)$ to represent the subset of $\mathcal{C}_{\mathrm{all-st-cuts}}(G)$ for which $s$, $u$, and $v$ are all on the same side. Finally, by $\mathcal{C}_\emptyset(G)$ we refer to the subset of $\mathcal{C}_{\mathrm{all-st-cuts}}(G)$ for which $t$, $u$, and $v$ are all on the same side.

Let $C_u, C_v, C_{u,v}$, and $C_\emptyset$ be the min cuts in $\mathcal{C}_u(G), \mathcal{C}_v(G), \mathcal{C}_{u,v}(G)$, and $\mathcal{C}_\emptyset(G)$, respectively, before $X_{s,u}, X_{s,v}, X_{t,u}$ and $X_{t,v}$ are sampled. It is an easy observation that sampling $X_{s,u}, X_{s,v}, X_{t,u}$ and $X_{t,v}$ and altering the weight of the edge $e$ changes the weight of all the cuts in $\mathcal{C}_u(G)$ by the same amount. This same observation also holds for $\mathcal{C}_v(G), \mathcal{C}_{u,v}(G)$ and $\mathcal{C}_\emptyset(G)$. This further implies that the min $s$-$t$ cut of $G$ after $X_{s,u}, X_{s,v}, X_{t,u}$ and $X_{t,v}$ are sampled will be among $C_u, C_v, C_{u,v}$, and $C_\emptyset$. Observe that these four cuts are also the minimum-weight cuts in $\mathcal{C}_u(G'), \mathcal{C}_v(G'), \mathcal{C}_{u,v}(G')$, and $\mathcal{C}_\emptyset(G')$, respectively.

In other words, the min $s$-$t$ cuts in $G$ and in $G'$ are among $C_u$, $C_v$, $C_{u,v}$ and $C_\emptyset$, but not necessarily the same; this holds both before and after sampling $X_{s,u}$, $X_{s,v}$, $X_{t,u}$ and $X_{t,v}$. In the rest of this proof, we show that sampling $X_{s,u}$, $X_{s,v}$, $X_{t,u}$ and $X_{t,v}$ makes it likely that the min $s$-$t$ cuts in $G$ and $G'$ are the same cut.

**The ratio of probabilities in $G'$ and $G$ that $C_u$ is the min $s$-$t$ cut.** Simply, our goal is to compute

$$\frac{\Pr\left[w_{G'}(C_u) < w_{G'}(C_v) \ \wedge \ w_{G'}(C_u) < w_{G'}(C_{u,v}) \ \wedge \ w_{G'}(C_u) < w_{G'}(C_\emptyset)\right]}{\Pr\left[w_G(C_u) < w_G(C_v) \ \wedge \ w_G(C_u) < w_G(C_{u,v}) \ \wedge \ w_G(C_u) < w_G(C_\emptyset)\right]}. \tag{1}$$

First consider the case $w_G(e) = w_{G'}(e) + \tau$, for some $\tau > 0$. For the ease of calculation, for $x \in \{\emptyset, \{u,v\}\}$, define $\Delta_x = w_G(C_u) - w_G(C_x) - \tau$ and define $\Delta_v = w_G(C_u) - w_G(C_v)$ *before* $X_{s,u}$, $X_{s,v}$, $X_{t,u}$ and $X_{t,v}$ are sampled. In the rest of this proof, we are analyzing the behavior of cuts when $X_{s,u}$, $X_{s,v}$, $X_{t,u}$ and $X_{t,v}$ **are sampled**. Then, we have

$$\Pr\left[w_G(C_u) < w_G(C_v) \ \wedge \ w_G(C_u) < w_G(C_{u,v}) \ \wedge \ w_G(C_u) < w_G(C_\emptyset)\right]$$
$$= \Pr[\Delta_v + X_{s,v} + X_{t,u} < X_{t,v} + X_{s,u}$$
$$\wedge \ \Delta_{u,v} + X_{s,v} + X_{t,u} + \tau < X_{t,v} + X_{t,u} \ \wedge \ \Delta_\emptyset + X_{s,v} + X_{t,u} + \tau < X_{s,v} + X_{s,u}]$$
$$= \Pr[\Delta_v + X_{s,v} + X_{t,u} < X_{t,v} + X_{s,u}$$
$$\wedge \ \Delta_{u,v} + X_{s,v} + \tau < X_{t,v} \ \wedge \ \Delta_\emptyset + X_{t,u} + \tau < X_{s,u}]. \tag{2}$$

Now, we replace $X_{s,v} - X_{t,v}$ by $x$ and $X_{t,u} - X_{s,u}$ by $y$. Then, Eq. (2) can be rewritten as

$$\Pr\left[w_G(C_u) < w_G(C_v) \ \wedge \ w_G(C_u) < w_G(C_{u,v}) \ \wedge \ w_G(C_u) < w_G(C_\emptyset)\right]$$
$$= \Pr\left[x + y < -\Delta_v \ \wedge \ x < -\Delta_{u,v} - \tau \ \wedge \ y < -\Delta_\emptyset - \tau\right].$$

Applying the same analysis for $G'$ we derive

$$\Pr\left[w_{G'}(C_u) < w_{G'}(C_v) \ \wedge \ w_{G'}(C_u) < w_{G'}(C_{u,v}) \ \wedge \ w_{G'}(C_u) < w_{G'}(C_\emptyset)\right]$$
$$= \Pr[\Delta_v + X_{s,v} + X_{t,u} < X_{t,v} + X_{s,u}$$
$$\wedge \ \Delta_{u,v} + X_{s,v} + X_{t,u} < X_{t,v} + X_{t,u} \ \wedge \ \Delta_\emptyset + X_{s,v} + X_{t,u} < X_{s,v} + X_{s,u}]$$
$$= \Pr[\Delta_v + X_{s,v} + X_{t,u} < X_{t,v} + X_{s,u} \ \wedge \ \Delta_{u,v} + X_{s,v} < X_{t,v} \ \wedge \ \Delta_\emptyset + X_{t,u} < X_{s,u}].$$

Replacing $X_{s,v} - X_{t,v}$ by $x$ and $X_{t,u} - X_{s,u}$ by $y$ yields

$$\Pr\left[w_{G'}(C_u) < w_{G'}(C_v) \ \wedge \ w_{G'}(C_u) < w_{G'}(C_{u,v}) \ \wedge \ w_{G'}(C_u) < w_{G'}(C_\emptyset)\right]$$
$$= \Pr\left[x + y < -\Delta_v \ \wedge \ x < -\Delta_{u,v} \ \wedge \ y < -\Delta_\emptyset\right].$$

Note that by Fact 2.3 we have that $x$ and $y$ follow $\mathrm{Lap}(1/\epsilon)$. Moreover, observe that the random variables $x$ and $y$ are independent by definition. Therefore, by invoking Lemma 3.1 with $\alpha = -\Delta_{u,v} - \tau$, $\beta = -\Delta_\emptyset - \tau$, and $\gamma = -\Delta_v$ we derive

$$\frac{\Pr\left[w_{G'}(C_u) < w_{G'}(C_v) \ \wedge \ w_{G'}(C_u) < w_{G'}(C_{u,v}) \ \wedge \ w_{G'}(C_u) < w_{G'}(C_\emptyset)\right]}{\Pr\left[w_G(C_u) < w_G(C_v) \ \wedge \ w_G(C_u) < w_G(C_{u,v}) \ \wedge \ w_G(C_u) < w_G(C_\emptyset)\right]} \le e^{4\tau\epsilon}.$$

Considering the second case $w_G(e) + \tau = w_{G'}(e)$, for some $\tau > 0$, we derive that the ratio Eq. (1) is upper-bounded by $1 \le e^{4\tau\epsilon}$.

**The remaining cases.** The proof for the remaining case, i.e., analysis when $C_v, C_{u,v}$ and $C_\emptyset$ is the minimum, follows the same steps as the case we have just analyzed.

**Finalizing the proof.** It remains to discuss two properties that are needed to complete the proof. First, our analysis above is applied for all but $4$ random variables $X$ fixed. Nevertheless, our analysis does not depend on how those random variables are fixed. Therefore, for any fixed cut $C$, summing/integrating over all the possible outcomes of those variables yields

$$\frac{\Pr\left[\text{Algorithm 1 outputs } C \text{ given } G\right]}{\Pr\left[\text{Algorithm 1 outputs } C \text{ given } G'\right]} \le e^{4\tau\epsilon}.$$

Second, our proof shows the ratio of a cut $C$ being the minimum one in $G$ and $G'$. However, the DP definition applies to any set of multiple cuts. It is folklore that in the case of pure DP, i.e., when $\delta = 0$, these two cases are equivalent. This completes the analysis. $\qquad\square$

## 3.2 Approximation Analysis

Observe that showing that Algorithm 1 has $O(n/\epsilon \cdot \log n)$ additive error is straightforward as a random variable drawn from $\mathrm{Exp}(b)$ is upper-bounded by $O(\log n/b)$ whp. Lemma 3.3 proves the $O(n/\epsilon)$ bound. On a very high level, our proof relies on the fact that for a sufficiently large constant $c$, it holds that only *a small fraction* of random variables $X_{i,j}$ exceeds $c/\epsilon$; this fraction is $e^{-c}$.

**Lemma 3.3.** *With probability at least $1 - n^{-2}$, Algorithm 1 outputs a min s-t cut with additive error $O(n/\epsilon)$.*

*Proof.* Let $G$ be an input graph to Algorithm 1 and let $\widetilde{G}$ be the graph after the edges on Line 2 are added. $\widetilde{G}$ contains $2(n-1)$ more edges than $G$. This proof shows that with probability at least $1 - n^{-2}$, the total sum of weights of all these edges is $O(n/\epsilon)$.

We first provide a brief intuition. For the sake of it, assume $\epsilon = 1$. Whp, each $X_{i,j}$ weighs at most $5 \log n$. Also, consider only those $X_{i,j}$ such that $X_{i,j} \geq 2$; the total sum of those random variables having weight less than 2 is $O(n)$. It is instructive to think of the interval $[2, 5 \log n]$ being partitioned into buckets of the form $[2^i, 2^{i+1})$. Then, the value of each edge added by Algorithm 1 falls into one of the buckets. Now, the task becomes upper-bounding the number $c_i$ of edges in bucket $i$. That is, we let $Y_{s,u}^i = 1$ iff $X_{s,u} \in [2^i, 2^{i+1})$, which results in $c_i = \sum_{u \in V}(Y_{s,u}^i + Y_{t,u}^i)$. Hence, $c_i$ is a sum of 0/1 independent random variables, and we can use Chernoff bound to argue about its concentration.

There are two cases. If $\mathbb{E}[c_i]$ is more than $O(\log n)$, then $c_i \in O(\mathbb{E}[c_i])$ by the Chernoff bound. If $\mathbb{E}[c_i] \in o(\log n)$, e.g., $\mathbb{E}[c_i] = O(1)$, we can not say that with high probability $c_i \in O(\mathbb{E}[c_i])$. Nevertheless, it still holds $c_i \in O(\log n)$ whp.

**Edges in $E(\widetilde{G}) \setminus E(G)$ with weights more than $5\frac{\log n}{\epsilon}$.** Let $Y \sim \mathrm{Exp}(\epsilon)$. By Fact 2.1, $\Pr\left[Y > 5\frac{\log n}{\epsilon}\right] = \exp(-5\log n) = n^{-5}$. Since each $X_{s,v}$ and $X_{t,v}$ is drawn from $\mathrm{Exp}(\epsilon)$, we have that for $n \geq 2$ with probability at least $1 - n^{-3}$ each of the edges in $E(\widetilde{G}) \setminus E(G)$ has weight at most $5\frac{\log n}{\epsilon}$.

**Edges in $E(\widetilde{G}) \setminus E(G)$ with weights at most $5\frac{\log n}{\epsilon}$.** Observe that the sum of the weights of all the edges in $E(\widetilde{G}) \setminus E(G)$ having weight at most $2/\epsilon$ is $O(n/\epsilon)$. Hence, we focus on the edge weights in the interval $\left[\frac{2}{\epsilon}, 5\frac{\log n}{\epsilon}\right]$. We partition this interval into $O(\log \log n)$ subintervals where each, except potentially the last one, of the form $[2^i/\epsilon, 2^{i+1}/\epsilon)$.

Let $Y \sim \mathrm{Exp}(\epsilon)$. We have

$$\Pr\left[Y \in [2^i/\epsilon, 2^{i+1}/\epsilon)\right] \leq \Pr\left[Y \geq 2^i/\epsilon\right] = e^{-2^i}.$$

Let $c_i$ be the number of random variables among $X_{s,v}$ and $X_{t,v}$ whose values belong to $[2^i/\epsilon, 2^{i+1}/\epsilon)$. Then, we derive

$$\mathbb{E}[c_i] \leq \frac{2(n-1)}{e^{2^i}} \leq \frac{2n}{2^{2^i}} \leq \frac{2n}{2^{2i}},$$

where to obtain the inequalities, we use that $i \geq 1$. By the Chernoff bound, for appropriately set constant $b > 0$, it holds that $c_i \leq b \cdot \max\left(\log n, \frac{2n}{2^{2i}}\right)$ with probability at least $1 - n^{-5}$. By the union bound, this claim holds for all the $O(\log \log n)$ partitions simultaneously and with probability at least $1 - n^{-4}$. Hence, with probability at least $1 - n^{-4}$, the sum of the edge-weights in $E(\widetilde{G}) \setminus E(G)$ across all the $O(\log \log n)$ partitions is at most

$$\sum_{i=1}^{\log(5 \log n)} b \cdot \max\left(\log n, \frac{2n}{2^{2i}}\right) \cdot \frac{2^{i+1}}{\epsilon} \leq \frac{O(\mathrm{poly} \, \log n)}{\epsilon} + 2b \sum_{i \geq 0} \frac{2n}{2^i \cdot \epsilon} = \mathrm{poly} \, \log n + 8b\frac{n}{\epsilon} \in O\left(\frac{n}{\epsilon}\right),$$

where we used $\max(\log n, 2n/2^{2i}) \leq \log n + 2n/2^{2i}$. By taking the union bound over both cases, we have that with probability at least $1 - n^{-2}$ it holds that the sum of the weights of all edges added to $G$ is $O(n/\epsilon)$. Hence, with probability $1 - n^{-2}$ at least, the min s-t cut in $\widetilde{G}$ has weight at most the min s-t cut in $G$ plus $O(n/\epsilon)$. This completes our analysis. $\qquad\square$

# 4 DP algorithm for multiway cut

In this section we show our approach for proving Theorem 1.3 restated below.

**Theorem 1.3.** *For any $\epsilon > 0$, there exists an $(\epsilon, 0)$-private algorithm for multiway $k$-cut on weighted undirected graphs that with probability $1 - 1/n$ at least returns a solution with value $2 \cdot OPT + O(n \log k/\epsilon)$, where $OPT$ is the value of the optimal non-private multiway $k$-cut.*

We first develop an algorithm for the non-private multiway $k$-cut problem and then use it to prove Theorem 1.3. The algorithm we design invokes a min $s$-$t$ cut procedure $O(\log k)$ times.

## 4.1 Solving Multiway Cut in $\log k$ Rounds of Min $s$-$t$ Cut

Our new myltiway $k$-cut algorithm is presented in Algorithm 2. Algorithm 2 first finds a cut that separates $s_1, \ldots, s_{k'}$ from $s_{k'+1}, \ldots, s_k$. This separation is obtained by contracting $s_1, \ldots, s_{k'}$ into a single node called $s$, contracting $s_{k'+1}, \ldots, s_k$ into a single node called $t$, and then running min $s$-$t$ cut on this new graph. Afterward, each of the two partitions is processed separately by recursion, and the algorithm outputs the union of the outputs on each of the two partitions.

---

**Algorithm 2:** Given a weighted graph $G$, this algorithm outputs a multiway $k$-cut.

---

**Input** : A weighted graph $G = (V, E, w)$
    Terminals $s_1, \ldots, s_k$
    A min $s$-$t$ cut algorithm $\mathcal{A}$ that with probability $1 - \alpha$ is $(1, e(n))$-approximate

1 **if** $k = 1$ **then**
2   **return** $G$ as one partition
3 **else**
4   Let $k' = \lfloor \frac{k}{2} \rfloor$.
5   Let $\widetilde{G}$ be the graph obtained by contracting $s_1, \ldots, s_{k'}$ into $s$ and contracting $s_{k'+1}, \ldots, s_k$
   into $t$.
6   Run algorithm $\mathcal{A}$ on $\widetilde{G}$ with terminals $s, t$ to obtain cut $C$ with partitions $\widetilde{G}_1$ and $\widetilde{G}_2$.
7   For $i = 1, 2$, let $G_i$ be the graph obtained from $\widetilde{G}_i$ by reversing the contraction of terminal
   nodes.
8   Let $C_1$ be the output of Algorithm 2 on $G_1$ and $C_2$ be the output of Algorithm 2 on $G_2$.
9   **return** $C_1 \cup C_2 \cup C$.

---

We first show that in each recursion level of the algorithm, we can run the min $s$-$t$ cut step (Line 6) of all the instances in that level *together* as one min $s$-$t$ cut, so that we only run $O(\log k)$ many min $s$-$t$ cuts.

**Lemma 4.1.** *Algorithm 2 is a reduction of multiway $k$-cut on $n$-node graphs to $O(\log k)$ many instances of min $s$-$t$ cut on $O(n)$-node graphs. Moreover, if $T(\mathcal{A}, n, m)$ is the running time of $\mathcal{A}$ on an $n$-node $m$-edge graphs, Algorithm 2 runs in $O(\log k \cdot [T(\mathcal{A}, n) + m])$.*

*Proof.* Let $G$ be the input graph and suppose that it has $n$ nodes. If we consider the recursion tree of the algorithm on $G$, we can perform the min $s$-$t$ cut step (Line 6) of *all of the subproblems on one level* of the recursion tree by a single min $s$-$t$ cut invocation. To see this, assume that $G_1^{r'}, \ldots, G_r^{r'}$ are in level $r'$ of the recursion tree for $r = 2^{r'-1}$, and for sake of simplicity, assume that $k$ is a power of 2. Let $s_i, t_i$ be the two terminals in $\widetilde{G}_i^{r'}$ (defined in Line 5) for $i \in \{1, \ldots, r\}$. We can think of the collection of graphs $\widetilde{G}_1^{r'}, \ldots, \widetilde{G}_r^{r'}$ as one graph $G^{r'}$. Contract $s_1, \ldots, s_r$ into $s$, and contract $t_1, \ldots, t_r$ into $t$, to obtain the graph $\widetilde{G}^{r'}$ from $G^{r'}$. Then performing a min $s$-$t$ cut algorithm on $\widetilde{G}^{r'}$ is equivalent to performing min $s$-$t$ cut on each of $G_i^{r'}$, as there is no edge between $G_i^{r'}$ and $G_j^{r'}$ for any $i, j$. Note that since $G_1^{r'}, \ldots, G_r^{r'}$ are disjoint and are subgraphs of $G$, $\widetilde{G}^{r'}$ has at most $n$ nodes. Moreover, the node contraction processes in each recursion level takes at most $O(m)$ as each edge is scanned at most once. $\square$

To prove that Algorithm 2 outputs a multiway $k$-cut that is a 2-approximation of the optimal multiway $k$-cut, we first present a few definitions. Given graph $G$ with node set $V$, a *partial* multiway $k$-cut is a set of disjoint node subsets $V_1, \ldots, V_k$ such that for each $i$, terminal $s_i \in V_i$. Note that $V_1 \cup \ldots \cup V_k$ is not necessarily the whole $V$. For a set of nodes $S \subseteq V$, let $\delta_G(S)$ be the sum of the weights of the edges with exactly one endpoint in $S$, i.e., the sum of the weights of the edges in $E(S, V \setminus S)$. Let $w(E(S))$ be the sum of the weights of the edges with both endpoints in $S$. Theorem 4.1 is the main result in this section and we state the proof of it the Supplementary material

**Theorem 4.1.** *Let $e(n)$ be a convex function and $G = (V, E, w)$ be a weighted graph. Let $\mathcal{A}$ be an algorithm that with probability $1 - \alpha$ outputs a $(1, e(n))$-approximate min $s$-$t$ cut for $e(n) = cn/\epsilon$ for constants $\epsilon > 0$, $c \geq 0$. Then, for any graph with terminals $s_1, \ldots, s_k$ and any partial multiway $k$-cut $V_1, \ldots, V_k$ of it, Algorithm 2 outputs a multiway cut that with probability $1 - \alpha O(\log k)$ has a value at most $\sum_{i=1}^{k} \delta(V_i) + w(E(\overline{V})) + O(\log(k) e(n))$ where $\overline{V} = V \setminus [\cup_{i=1}^{k} V_i]$.*

A direct Corollary of Theorem 4.1 is a 2-approximation algorithm of the optimal multiway $k$-cut. Recall that the novelty of this algorithm is in using $O(\log k)$ many min $s$-$t$ cut runs as opposed to $O(k)$ which is depicted in Lemma 4.1.

**Corollary 4.1.** *Given a graph $G$, integer $k \geq 2$ and any exact min $s$-$t$ cut algorithm, Algorithm 2 returns a multiway $k$-cut that is a 2-approximation of the optimal multiway $k$-cut.*

*Proof.* Let $\mathcal{A}$ be an exact $s$-$t$-cut algorithm that is part of the input to Algorithm 2. So we have $e(n) = 0$. Let $C_{ALG}$ be the output of Algorithm 2. By Theorem 4.1, $C_{ALG}$ is a multiway $k$-cut. Let $C_{OPT}$ be the optimal multiway $k$-cut with partitions $V_1, \ldots, V_k$. Note that there are no nodes that are not partitioned, and hence $w(E(\overline{V})) = 0$. So by Theorem 4.1, we have that $w(C_{ALG}) \leq \sum_{i=1}^{k} \delta_G(V_i) = 2w(C_{OPT})$. □

## 4.2 Differentially Private Multiway Cut

Our Differentially Private Multiway cut algorithm is essentially Algorithm 2 where we use a differentially private min $s$-$t$ cut algorithm (such as Algorithm 1) in Line 6 to compute a differentially private multiway $k$-cut. This concludes the approach for Theorem 1.3 which is a direct corollary of Theorem 4.2 below.

**Theorem 4.2.** *Let $\epsilon > 0$ be a privacy parameter, $G$ an $n$-node graph, and $\mathcal{A}$ an $\epsilon$-DP algorithm that with probability at least $1 - 1/n^2$ computes a min $s$-$t$ cut with additive error $O(e(n)/\epsilon)$ on any $n$-node graph. Then, Algorithm 2 is a $(\epsilon \log k)$-DP algorithm that with probability at least $1 - O(\log k) \cdot n^{-2}$ finds a multiway $k$-cut with multiplicative error $2$ and additive error $O(\log k \cdot e(n)/\epsilon)$.*

*Proof.* The error guarantees are a direct result of Theorem 4.1 and Lemma 3.3. Hence here we prove the privacy guarantee. Consider the recursion tree of Algorithm 2 which has depth $O(\log k)$. As proved by Lemma 4.1, each level of the recursion tree consists of running algorithm $\mathcal{A}$ on disjoint graphs, and so each level is $\epsilon$-DP. By basic composition (see Definition 2.5), since Algorithm 2 is a composition of $O(\log k)$ many $\epsilon$-DP mechanisms, it is $(\epsilon \log k)$-DP. □

# 5 Empirical Evaluation

We perform an empirical evaluation on the additive error of Algorithm 1 and validate our theoretical bounds.

**Set-up.** As our base graph, we use email-Eu-core network [22] which is an undirected unweighted $1,005$-node $25,571$-edge graph. This graph represents email exchanges, where two nodes $u$ and $v$ are adjacent if the person representing $u$ has sent at least one email to the person representing $v$ or vice-versa. However, this graph does not account for multiple emails sent between two individuals. To make the graph more realistic, we add random weights to the edges from the exponential distribution with a mean of $40$ (rounded to an integer) to denote the number of emails sent between two nodes. We take $10$ percent of the nodes and contract them into a terminal node $s$, and take another $10$ percent of the nodes and contract them into another terminal node $t$. We make different instances of the problem by choosing the nodes that contract into $s$ or $t$ uniformly at random. Note that node contraction into terminals is a standard practice in real-world min $s$-$t$ cut instances, as there are often nodes with

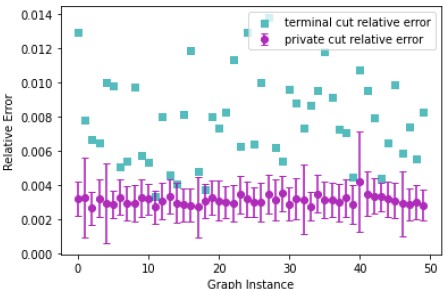 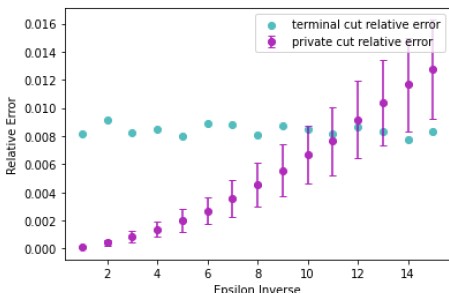

Figure 1: (left) Terminal cut and private cut with error bars over 50 graph instances, for $\epsilon = 0.5$. For each instance, Algorithm 1 was run 100 times. (right) Average terminal cut value and private cut value over different values of $\epsilon$. The $x$ axis is $\frac{1}{\epsilon}$. For each value of $\epsilon$, the set up is the same as in the left figure.

predetermined partitions and to make a $s$-$t$ cut (or multiway $k$-cut) instance one needs to contract these nodes into one terminal node for each partition. We take the floor of the values $X_{s,u}$ and $X_{t,u}$ to obtain integral weights.

**Baseline.** Let $C_s$ be the cut where one partition consists of only $s$, and let $C_t$ be the cut where one partition consists of only $t$. We compare the cut value output by Algorithm 1 against $\min(w(C_s), w(C_t))$. Particularly, if $C_0$ is the min $s$-$t$ cut of an instance and Algorithm 1 outputs $C_{alg}$, then we compare relative errors $\frac{w(C_{alg}) - w(C_0)}{w(C_0)}$ and $\frac{\min(w(C_s), w(C_t)) - w(C_0)}{w(C_0)}$*. We refer to the former value as *the private cut relative error*, and to the later value as *the terminal cut relative error*.

**Results.** We first set $\epsilon = 0.5$. We then consider 50 graph instances, and for each of them, we evaluate the average private error over 50 rounds of randomness used in Algorithm 1, see Figure 1(left) for the results. For almost all instances, the private cut error (including the error bars) is less than the terminal cut error. Moreover, the private cut error is quite stable, and far below the theoretical bound of $n/\epsilon$. We next change $\epsilon$ and repeat the set-up above for each $\epsilon \in \{\frac{1}{15}, \frac{1}{14}, \ldots, \frac{1}{2}, 1\}$. For each $\epsilon$ in this set we produce 50 graph instances, measure the average private error over 100 rounds of randomness for each, and then average the terminal cut error and mean private cut error for each graph instance. In this way, we obtain an average value for terminal cut and private cut errors for each of the $\epsilon$ values; we refer to Figure 1(right) for the results. The linear relationship between the private cut error and $\frac{1}{\epsilon}$ can be observed in Figure 1(right).

# 6 Conclusion And Future Work

In this work, we investigate the differential privacy of the min $s$-$t$ cut and of, its generalization, the multiway $k$-cut problem. For the first problem, we show almost tight lower- and upper-bounds in terms of the error guarantee. Moreover, our DP algorithm is as fast as the fastest non-private one. For the multiway $k$-cut problem, we develop a novel approach that yields only $O(n/\epsilon \cdot \log k)$ additive error, as opposed to $O(n/\epsilon \cdot k)$ that standard algorithms lead to, in order to achieve DP. This algorithm might be of independent interest. In particular, the *total* running time of our multiway $k$-cut algorithm is only $O(\log k)$ more than that of the fastest non-private min $s$-$t$ cut. This is in stark contrast with the prior non-private 2-approximate approach, which is $O(k)$ times slower than the fastest non-private min $s$-$t$ cut algorithm.

Comparing our lower- and upper-bounds, it remains to close the gap of $1/\epsilon$ multiplicative factor in the additive error of the min $s$-$t$ cut problem. It also remains open whether $O(\log k)$ factor in the additive error for the multiway $k$-cut problem is needed. We believe that techniques relying on solving LPs might be able to avoid this dependence on $O(\log k)$. Finally, it would be interesting to explore whether smaller additive error is possible for special but interesting classes of graphs.

---

*As another standard heuristic one could consider a random $s$-$t$ cut. We do not include this baseline here as it often has a very high error and the terminal cut does much better than a random cut.

## Acknowledgments and Disclosure of Funding

We are grateful to anonymous reviewers for the valuable feedback. S. Mitrović was supported by the Google Research Scholar Program.

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

# A  Multiple min $s$-$t$ cuts

Let $G$ and $G'$ be two neighboring graphs, and let $\widetilde{G}$ and $\widetilde{G}'$, respectively, be their modified versions constructed by Algorithm 1. Algorithm 1 outputs a min $s$-$t$ cut in $G$. However, what happens if there are multiple min $s$-$t$ cuts in $\widetilde{G}$ and the algorithm invoked on Line 4 breaks ties in a way that depends on whether a specific edge $e$ appears in $G$ or not? If it happens that $e$ is the edge difference between $G$ and $G'$, then such a tie-breaking rule might reveal additional information about $G$ and $G'$. We now outline how this can be bypassed.

Observe that if the random variables $X_{s,u}$ and $X_{t,u}$ were sampled by using infinite bit precision, then with probability 1 no two cuts would have the same value. So, consider a more realistic situation where edge-weights are represented by $O(\log n)$ bits, and assume that the least significant bit corresponds to the value $2^{-t}$, for an integer $t \geq 0$. We show how to modify edge-weights by additional $O(\log n)$ bits that have extremely small values but will help obtain a unique min $s$-$t$ cut. Our modification consists of two steps.

**First step.** All the bits corresponding to values from $2^{-t-1}$ to $2^{-t-2\log n}$ remain 0, while those corresponding to larger values remain unchanged. This is done so that even summing across all – but at most $\binom{n}{2}$ – edges it holds that no matter what the bits corresponding to values $2^{-t-2\log n-1}$ and less are, their total value is less than $2^{-t}$. Hence, if the weight of cut $C_1$ is smaller than the weight of cut $C_2$ before the modifications we undertake, then $C_1$ has a smaller weight than $C_2$ after the modifications as well.

**Second step.** We first recall the celebrated Isolation lemma.

**Lemma A.1** (Isolation lemma, [26])**.** *Let $T$ and $N$ be positive integers, and let $\mathcal{F}$ be an arbitrary nonempty family of subsets of the universe $\{1, \ldots, T\}$. Suppose each element $x \in \{1, \ldots, T\}$ in the universe receives an integer weight $g(x)$, each of which is chosen independently and uniformly at random from $\{1, \ldots, N\}$. The weight of a set $S \in \mathcal{F}$ is defined as $g(S) = \sum_{x \in S} g(x)$.*

*Then, with probability at least $1 - T/N$ there is a unique set in $\mathcal{F}$ that has the minimum weight among all sets of $\mathcal{F}$.*

We now apply Lemma A.1 to conclude our modification of the edge weights in $\widetilde{G}$. We let the universe $\{1, \ldots, T\}$ from that lemma be the following $2(n-2)$ elements $\mathcal{U} = \{(s, v) \mid v \in V(G) \setminus \{s, t\}\} \cup \{(t, v) \mid v \in V(G) \setminus \{s, t\}\}$. Then, we let $\mathcal{F}$ represent all min $s$-$t$ cuts in $\widetilde{G}$, i.e., $S \subseteq \mathcal{U}$ belongs to $\mathcal{F}$ iff there is a min $s$-$t$ cut $C$ in $\widetilde{G}$ such that for each $(a, b) \in S$ the cut $C$ contains $X_{a,b}$. So, by letting $N = 2n^2$, we derive that with probability $1 - 1/n$ it holds that no two cuts represented by $\mathcal{F}$ have the same minimum value **with respect to** $g$ defined in Lemma A.1. To implement $g$ in our modification of weights, we modify the bits of each $X_{s,v}$ and $X_{t,v}$ corresponding to values from $2^{-t-2\log n-1}$ to $2^{-t-2\log n-\log N}$ to be an integer between 1 and $N$ chosen uniformly at random.

Only after these modifications, we invoke Line 4 of Algorithm 1. Note that the family of cuts $\mathcal{F}$ is defined only for the sake of analysis. It is not needed to know it algorithmically.

# B  Lower Bound for min $s$-$t$ cut error

In this section, we prove our lower bound. Our high-level idea is similar to that of [8] for proving a lower bound for private algorithms for correlation clustering.

**Theorem 1.2.** *Any $(\epsilon, \delta)$-differential private algorithm for min $s$-$t$ cut on $n$-node graphs requires expected additive error of at least $n/20$ for any $\epsilon \leq 1$ and $\delta \leq 0.1$.*

*Proof.* For the sake of contradiction, let $\mathcal{A}$ be a $(\epsilon, \delta)$-differential private algorithm for min $s$-$t$ cut that on any input $n$-node graph outputs an $s$-$t$ cut that has expected additive error of less than $n/20$. We construct a set of $2^n$ graphs $S$ and show that $\mathcal{A}$ cannot have low expected cost on all of the graphs on this set while preserving privacy.

The node set of all the graphs in $S$ are the same and consist of $V = \{s, t, v_1, \ldots, v_n\}$ where $s$ and $t$ are the terminals of the graph and $n > 30$. For any $\tau \in \{0, 1\}^n$, let $G_\tau$ be the graph on node set $V$

with the following edges: For any $1 \le i \le n$, if $\tau_i = 1$, then there is an edge between $s$ and $v_i$. If $\tau = 0$, then there is an edge between $t$ and $v_i$. Note that $v_i$ is attached to exactly one of the terminals $s$ and $t$. Moreover, the min $s$-$t$ cut of each graph $G_\tau$ is zero.

Algorithm $\mathcal{A}$ determines for each $i$ if $v_i$ is on the $s$-side of the output cut or the $t$-side. The contribution of each node $v_i$ to the total error is the number of edges attached to $v_i$ that are in the cut. We denote this random variable in graph $G_\tau$ by $e_\tau(v_i)$. Since there are no edges between any two non-terminal nodes in any of the graphs $G_\tau$, the total error of the output is the sum of these individual errors, i.e., $\sum_{i=0}^{n} e_\tau(v_i)$. Let $\bar{e}_\tau(v_i)$ be the expected value of $e_\tau(v_i)$ over the outputs of $\mathcal{A}$ given $G_\tau$.

Let $p_\tau^{(i)}$ be the marginal probability that $v_i$ is on the $s$-side of the output $s$-$t$ cut in $G_\tau$. If $\tau_i = 0$, then $v_i$ is connected to $t$ and so $\bar{e}_\tau(v_i) = p_\tau^{(i)}$. If $\tau_i = 1$, then $v_i$ is connected to $s$ and so $\bar{e}_\tau(v_i) = 1 - p_\tau^{(i)}$. By the assumption that $\mathcal{A}$ has a low expected error on every input, we have that for any $\tau \in \{0,1\}^n$,

$$(n+2)/20 > \sum_{i, \tau_i = 0} p_\tau^{(i)} + \sum_{i, \tau_i = 1} (1 - p_\tau^{(i)}) \tag{3}$$

Let $S_i$ be the set of $\tau \in \{0,1\}^n$ such that $\tau_i = 0$, and $\bar{S}_i$ be the complement of $S_i$, so that $\tau \in \bar{S}_i$ if $\tau_i = 1$. Note that $|S_i| = |\bar{S}_i| = 2^{n-1}$. Fix some $i$, and for any $\tau \in \{0,1\}^n$, let $\tau'$ be the same as $\tau$ except for the $i$-th entry being different, i.e., for all $j \ne i$, $\tau_j = \tau'_j$, and $\tau_i \ne \tau'_i$. Since $G_\tau$ and $G_{\tau'}$ only differ in two edges, from $\mathcal{A}$ being $(\epsilon, \delta)$-differentially private for any $j$ we have $p_\tau^{(j)} \le e^{2\epsilon} \cdot p_{\tau'}^{(j)} + \delta$. So for any $i, j$ we have

$$\sum_{\tau \in S_i} p_\tau^{(j)} \le \sum_{\tau \in \bar{S}_i} (e^{2\epsilon} p_\tau^{(j)} + \delta) \tag{4}$$

From Eq. (3) we have

$$2^n \cdot 0.05(n+2) > \sum_{\tau \in \{0,1\}^n} \sum_{i : \tau_i = 1} (1 - p_\tau^{(i)})$$
$$= \sum_{i=1}^{n} \sum_{\tau \in S_i} (1 - p_\tau^{(i)})$$
$$\ge \sum_{i=1}^{n} \sum_{\tau \in \bar{S}_i} (1 - [e^{2\epsilon} p_\tau^{(i)} + \delta])$$
$$= n 2^{n-1}(1 - \delta) - e^{2\epsilon} \sum_{i=1}^{n} \sum_{\tau \in \bar{S}_i} p_\tau^{(i)}$$

Where the last inequality comes from Eq. (4). Using Eq. (3) again, we have that $\sum_{i=1}^{n} \sum_{\tau \in \bar{S}_i} p_\tau^{(i)} < 2^n \cdot 0.05(n+2)$, so we have that $2^n \cdot 0.05(n+2) > n 2^{n-1}(1-\delta) - e^{2\epsilon}(2^n \cdot 0.05(n+2))$. Dividing by $2^n$ we have

$$0.05(n+2)(1 + e^{2\epsilon}) > n(1 - \delta)/2.$$

Now since $\epsilon \le 1$, $\delta \le 0.1$, and $e^2 < 7.4$ we get that

$$0.05 \cdot 8.4(n+2) > 0.45n$$

Hence we have $n < 28$ which is a contradiction to $n > 30$. $\qquad\square$

## C   Omitted Proofs

### C.1   Proof of Claim 1

By definition, we have

$$\frac{f_{\text{Lap}}(t+\tau)}{f_{\text{Lap}}(t)} = \frac{\frac{\epsilon}{2} \exp(-\epsilon|t+\tau|)}{\frac{\epsilon}{2} \exp(-\epsilon|t|)} = \exp(-\epsilon|t+\tau| + \epsilon|t|) \le \exp(\tau\epsilon). \tag{5}$$

Also by definition, it holds $F_{\text{Lap}}(t+\tau) = \int_{-\infty}^{t+\tau} f_{\text{Lap}}(x)dx$. Using Eq. (5) we derive

$$F_{\text{Lap}}(t+\tau) \le \exp(\tau\epsilon) \int_{-\infty}^{t+\tau} f_{\text{Lap}}(x-\tau)dx = \exp(\tau\epsilon) \int_{-\infty}^{t} f_{\text{Lap}}(x)dx = \exp(\tau\epsilon) F_{\text{Lap}}(t).$$

## C.2 Proof of Lemma 3.1

To prove the lower-bound, we observe that if $x < \alpha$ and $y < \beta$, then $x < \alpha + \tau$ and $y < \beta + \tau$ as well. Hence, it trivially holds $P(\alpha + \tau, \beta + \tau, \gamma) \geq P(\alpha, \beta, \gamma)$ and hence

$$\frac{P(\alpha + \tau, \beta + \tau, \gamma)}{P(\alpha, \beta, \gamma)} \geq 1.$$

We now analyze the upper-bound. For the sake of brevity, in the rest of this proof, we use $F$ to denote $F_{\mathrm{Lap}}$ and $f$ to denote $f_{\mathrm{Lap}}$. We consider three cases depending on parameters $\alpha, \beta, \gamma$.

**Case $\gamma \geq \alpha + \beta + 2\tau$.** In this case we have $\Pr[x + y < \gamma | x < \alpha, y < \beta] = 1 = \Pr[x + y < \gamma | x < \alpha + \tau, y < \beta + \tau]$. So we have that

$$\begin{aligned}
P(\alpha, \beta, \gamma) &= \Pr[x + y < \gamma | x < \alpha, y < \beta] \cdot \Pr[x < \alpha, y < \beta] \\
&= \Pr[x < \alpha, y < \beta] \\
&= F(\alpha)F(\beta)
\end{aligned}$$

Similarly, $P(\alpha + \tau, \beta + \tau, \gamma) = F(\alpha + \tau)F(\beta + \tau)$. Now using Claim 1, we obtain that $\frac{P(\alpha+\tau,\beta+\tau,\gamma)}{P(\alpha,\beta,\gamma)} \leq e^{2\tau\epsilon}$.

**Case $\gamma < \alpha + \beta$.** We write $P(\alpha, \beta, \gamma)$ as follows.

$$\begin{aligned}
P(\alpha, \beta, \gamma) &= \int_{-\infty}^{\beta} \int_{-\infty}^{\min(\alpha, \gamma - y)} f_x(x|y) dx \, f_y(y) dy \\
&= \int_{-\infty}^{\beta} F(\min(\alpha, \gamma - y)) f(y) dy \\
&= F(\alpha) \int_{-\infty}^{\gamma - \alpha} f(y) dy + \int_{\gamma - \alpha}^{\beta} F(\gamma - y) f(y) dy \\
&= F(\alpha) F(\gamma - \alpha) + \int_{\gamma - \alpha}^{\beta} F(\gamma - y) f(y) dy \tag{6}
\end{aligned}$$

Similar to Eq. (6) we have

$$P(\alpha + \tau, \beta + \tau, \gamma) = F(\alpha + \tau) F(\gamma - \alpha - \tau) + \int_{\gamma - \alpha - \tau}^{\beta + \tau} F(\gamma - y) f(y) dy \tag{7}$$

Now we rewrite Eq. (6) as follows to obtain a lower bound on $P(\alpha, \beta, \gamma)$.

$$\begin{aligned}
P(\alpha, \beta, \gamma) &= F(\alpha) F(\gamma - \alpha - 2\tau) + \int_{\gamma - \alpha - 2\tau}^{\gamma - \alpha} F(\alpha) f(y) dy + \int_{\gamma - \alpha}^{\beta} F(\gamma - y) f(y) dy \\
&\geq F(\alpha) F(\gamma - \alpha - 2\tau) + e^{-2\tau\epsilon} \int_{\gamma - \alpha - 2\tau}^{\beta} F(\gamma - y) f(y) dy \tag{8}
\end{aligned}$$

In obtaining the inequality, we used the fact that if $y \in [\gamma - \alpha - 2\tau, \gamma - \alpha]$ then $0 \leq (\gamma - y) - \alpha \leq 2\tau$ and so by Claim 1 we have $F(\alpha) \geq e^{-2\tau\epsilon} F(\gamma - y)$.

Now we compare the two terms of Eq. (8) with Eq. (7). By Claim 1 we have that $F(\alpha)F(\gamma - \alpha - 2\tau) \geq e^{-2\tau\epsilon} F(\alpha + \tau) F(\gamma - \alpha - \tau)$ and $\int_{\gamma - \alpha - 2\tau}^{\beta} F(\gamma - y) f(y) dy \geq e^{-\tau\epsilon} \int_{\gamma - \alpha - \tau}^{\beta + \tau} F(\gamma - y) f(y) dy$. So we have $P(\alpha, \beta, \gamma) \geq e^{-3\tau\epsilon} P(\alpha + \tau, \beta + \tau, \gamma)$.

**Case** $\alpha + \beta \leq \gamma < \alpha + \beta + 2\tau$. Then

$$P(\alpha, \beta, \gamma) = \int_{-\infty}^{\beta} \int_{-\infty}^{\min(\alpha, \gamma - y)} f_x(x|y) dx f_y(y) dy$$

$$= \int_{-\infty}^{\beta} F(\min(\alpha, \gamma - y)) f(y) dy$$

$$= F(\alpha) F(\beta) \tag{9}$$

$$\geq e^{-2\tau\epsilon} F(\alpha + \tau) F(\beta + \tau) \tag{10}$$

$$= e^{-2\tau\epsilon} (F(\alpha + \tau) F(\beta - \tau) + F(\alpha + \tau)(F(\beta + \tau) - F(\beta - \tau)))$$

$$\geq e^{-4\tau\epsilon} (F(\alpha + \tau) F(\beta + \tau) + F(\alpha + \tau)(F(\beta + \tau) - F(\beta - \tau))). \tag{11}$$

Note that Eq. (9) is obtained since for any $y \leq \beta$ we have $\alpha \leq \gamma - y$. Eq. (10) and Eq. (11) are both obtained using Claim 1. One can easily verify that Eq. (7) for $P(\alpha + 1, \beta + 1, \gamma)$ holds in this case as well. Using the fact that $\gamma - \alpha - \tau \leq \beta + \gamma$ and $F$ being a non-decreasing function, we further lower-bound Eq. (7) as

$$P(\alpha + \tau, \beta + \tau, \gamma) \leq F(\alpha + \tau) F(\beta + \tau) + \int_{\gamma - \alpha - \tau}^{\beta + \tau} F(\gamma - y) f(y) dy$$

$$\leq F(\alpha + \tau) F(\beta + \tau) + F(\alpha + \tau) \int_{\gamma - \alpha - \tau}^{\beta + \tau} f(y) dy$$

$$= F(\alpha + \tau) F(\beta + \tau) + F(\alpha + \tau)(F(\beta + \tau) - F(\gamma - \alpha - \tau))$$

$$\leq F(\alpha + \tau) F(\beta + \tau) + F(\alpha + \tau)(F(\beta + \tau) - F(\beta - \tau)). \tag{12}$$

Eqs. (11) and (12) conclude the analysis of this case as well.

### C.3   Proof of Theorem 4.1

We use induction on $k$ to prove the theorem. Suppose that Algorithm 2 outputs $C_{ALG}$. We show that the $C_{ALG}$ is a multiway $k$-cut and that the value of $C_{ALG}$ is at most $w(E(\overline{V})) + \sum_{i=1}^{k} \delta(V_i) + 2\log(k)e(n)$. We will first perform the analysis of approximation assuming that $\mathcal{A}$ provides the stated approximation deterministically, and at the end of this proof, we will take into account that the approximation guarantee holds with probability $1 - \alpha$.

**Base case:** $k = 1$.   If $k = 1$, then $C_{ALG} = \emptyset$, and so it is a multiway 1-cut and $w(C_{ALG}) = 0 \leq \delta(V_1) + w(E(\overline{V}))$.

**Inductive step:** $k \geq 2$.   So suppose that $k \geq 2$. Hence, $k' \geq 1$ and $k - k' \geq 1$, where $k'$ is defined on Line 4. Let $(A, B)$ be the $s$-$t$ cut obtained in Line 6, where $\widetilde{G}_1$ is the graph induced on $A$ and $\widetilde{G}_2$ is the graph induced on $B$. Since the only terminals in $\widetilde{G}_1$ are $s_1, \ldots, s_{k'}$, we have that $V_1 \cap A, \ldots, V_{k'} \cap A$ is a partial multiway $k'$-cut on $\widetilde{G}_1$. By the induction hypothesis, the cost of the multiway cut that Algorithm 2 finds on $\widetilde{G}_1$ is at most $w(E(\overline{V} \cap A)) + \sum_{i=1}^{k'} \delta_{\widetilde{G}_1}(V_i \cap A) + 2\log(k')e(|A|)$. Similarly, by considering the partial multiway $(k - k')$ cut $V_{k'+1} \cap B, \ldots, V_k \cap B$ on $\widetilde{G}_2$, the cost of the multiway cut that Algorithm 2 finds on $\widetilde{G}_2$ is at most $w(E(\overline{V} \cap B)) + \sum_{i=k'+1}^{k} \delta_{\widetilde{G}_2}(V_i \cap B) + 2\log(k - k')e(|B|)$. So the total cost $w(C_{ALG})$ of the multiway cut that Algorithm 2 outputs is at most

$$w(C_{ALG}) \leq w(E(\overline{V} \cap A)) + w(E(\overline{V} \cap B)) \tag{13}$$

$$+ \sum_{i=1}^{k'} \delta_{\widetilde{G}_1}(V_i \cap A) + \sum_{i=k'+1}^{k} \delta_{\widetilde{G}_2}(V_i \cap B)$$

$$+ w(E(A, B))$$

$$+ 2\log(k')e(|A|) + 2\log(k - k')e(|B|)$$

First note that $C_{ALG}$ is a multiway $k$-cut: this is because by induction the output of the algorithm on $\widetilde{G}_1$ is a multiway $k'$-cut and the output of the algorithm on $\widetilde{G}_2$ is a multiway $(k - k')$-cut. Moreover,

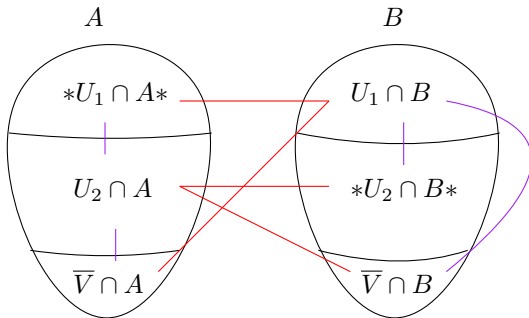

Figure 2: Node subsets of graph $G$. The subsets in asterisks have terminals in them. Red edges indicate the left-hand side edges in Eq. (14), and purple edges indicate the edges on the right-hand side in Eq. (14).

$E(A, B) \in C_{ALG}$. So the union of these cuts and $E(A, B)$ is a $k$-cut, and since each terminal is in exactly one partition, it is a multiway $k$-cut.

Now we prove the value guarantees. Let $U_1 = V_1 \cup \ldots \cup V_{k'}$ and $U_2 = V_{k'+1} \cup \ldots \cup V_k$. So $U = U_1 \cup U_2 = V_1 \cup \ldots \cup V_k$ is the set of nodes that are in at least one partition. Recall that $\overline{V} = V \setminus U$ is the set of nodes that are not in any partition.

Consider the following cut that separates $\{s_1, \ldots, s_{k'}\}$ from $\{s_{k'+1}, \ldots, s_k\}$: Let $A' = [U_1 \cap A] \cup [U_1 \cap B] \cup [\overline{V} \cap A]$. Let $B' = [U_2 \cap B] \cup [U_2 \cap A] \cup [\overline{V} \cap B]$. Since $(A, B)$ is a *min* cut that separates $\{s_1, \ldots, s_{k'}\}$ from $\{s_{k'+1}, \ldots, s_k\}$ with additive error $e(n)$, we have $w(E(A, B)) \leq w(E(A', B')) + e(n)$. Note that $A = [U_1 \cap A] \cup [U_2 \cap A] \cup [\overline{V} \cap A]$ and $B = [U_1 \cap B] \cup [U_2 \cap B] \cup [\overline{V} \cap B]$. So turning $(A, B)$ into $(A', B')$ is equivalent to switching $U_2 \cap A$ and $U_1 \cap B$ between $A$ and $B$. So we have that

$$w(E(U_2 \cap A, U_2 \cap B)) + w(E(U_2 \cap A, \overline{V} \cap B)) + w(E(U_1 \cap B, U_1 \cap A)) + w(E(U_1 \cap B, \overline{V} \cap A))$$
$$\leq \tag{14}$$
$$w(E(U_2 \cap A, U_1 \cap A)) + w(E(U_2 \cap A, \overline{V} \cap A)) + w(E(U_1 \cap B, U_2 \cap B)) + w(E(U_1 \cap B, \overline{V} \cap B))$$
$$+ e(n)$$

Eq. (14) is illustrated in Figure 2. Using Eq. (14), we obtain that

$$
\begin{aligned}
w(E(A, B)) &= w(E(U_2 \cap A, U_2 \cap B)) + w(E(U_2 \cap A, \overline{V} \cap B)) \\
&\quad + w(E(U_1 \cap B, U_1 \cap A)) + w(E(U_1 \cap B, \overline{V} \cap A)) \\
&\quad + w(E(U_2 \cap A, U_1 \cap B)) + w(E([U_1 \cap A] \cup [\overline{V} \cap A], [U_2 \cap B] \cup [\overline{V} \cap B])) \\
&\leq w(E(U_2 \cap A, U_1 \cap A)) + w(E(U_2 \cap A, \overline{V} \cap A)) \\
&\quad + w(E(U_1 \cap B, U_2 \cap B)) + w(E(U_1 \cap B, \overline{V} \cap B)) \\
&\quad + w(E(U_2 \cap A, U_1 \cap B)) + w(E([U_1 \cap A] \cup [\overline{V} \cap A], [U_2 \cap B] \cup [\overline{V} \cap B])) \\
&\quad + e(n)
\end{aligned}
$$

So we conclude that

$$
\begin{aligned}
w(E(A, B)) \leq\ & w(E(U_1 \cap B, [U_2 \cap B] \cup [\overline{V} \cap B] \cup [U_2 \cap A])) \tag{15} \\
& + w(E(U_1 \cap A, [U_2 \cap B] \cup [\overline{V} \cap B])) \\
& + w(E(U_2 \cap A, [U_1 \cap A] \cup [\overline{V} \cap A])) \\
& + w(E(U_2 \cap B, \overline{V} \cap A)) \\
& + w(E(\overline{V} \cap A, \overline{V} \cap B)) \\
& + e(n)
\end{aligned}
$$

We substitute $w(E(A, B))$ in Eq. (13) using Eq. (15). Recall that $U_1 = \cup_{i=1}^{k'} V_i$, $\delta_{\widetilde{G}_1}(V_i \cap A) = w(E(V_i \cap A, A \setminus V_i))$ and $\delta_G(V_i) = w(E(V_i, V \setminus V_i))$. For any $i \in \{1, \ldots, k'\}$, we have that

$E(V_i \cap B, [U_2 \cap B] \cup [\overline{V} \cap B] \cup [U_2 \cap A])$ and $E(V_i \cap A, [U_2 \cap B] \cup [\overline{V} \cap B])$ are both disjoint from $E(V_i \cap A, A \setminus V_i)$. Moreover all these three terms appear in $E(V_i, V \setminus V_i)$. So we have

$$w(E(U_1 \cap B, [U_2 \cap B] \cup [\overline{V} \cap B] \cup [U_2 \cap A]))$$

$$+ w(E(U_1 \cap A, [U_2 \cap B] \cup [\overline{V} \cap B])) + \sum_{i=1}^{k'} \delta_{\widetilde{G}_1}(V_i \cap A)$$

$$\leq \sum_{i=1}^{k'} \delta_G(V_i)$$

Note that the first two terms above are the first two terms in Eq. (15). Similarly, we have

$$w(E(U_2 \cap A, [U_1 \cap A] \cup [\overline{V} \cap A])) + w(E(U_2 \cap B, \overline{V} \cap A)) + \sum_{i=k'+1}^{k} \delta_{\widetilde{G}_2}(V_i \cap B) \leq \sum_{i=k'+1}^{k} \delta_G(V_i) \quad (16)$$

Note that the first two terms above are the third and forth terms in Eq. (15). Finally $w(E(\overline{V} \cap A)) + w(E(\overline{V} \cap B)) + w(E(\overline{V} \cap A, \overline{V} \cap B)) \leq w(E(\overline{V}))$. So, we upper-bound Eq. (13) as

$$w(C_{ALG}) \leq \sum_{i=1}^{k} \delta_G(V_i) + w(E(\overline{V}))$$

$$+ e(n) + 2\log(k')e(|A|) + 2\log(k-k')e(|B|).$$

Since $k' = \lfloor k/2 \rfloor$ and $k - k' = \lceil k/2 \rceil$, we have that $k' \leq \lfloor \frac{k+1}{2} \rfloor$ and $k - k' \leq \lfloor \frac{k+1}{2} \rfloor$. Moreover, since $e = cn/\epsilon$ for $\epsilon > 0$ and $c \geq 0$, we have that $e(|A|) + e(|B|) \leq e(|A| + |B|) = e(n)$. Therefore,

$$e(n) + 2\log(k')e(|A|) + 2\log(k-k')e(|B|)$$

$$\leq e(n)\left(1 + 2\log\left(\left\lfloor \frac{k+1}{2} \right\rfloor\right)\right) \leq 2\log(k)e(n).$$

The above inequality finishes the approximation proof.

**The success probability.** As proved by Lemma 4.1, the min $s$-$t$ cut computations by Algorithm 2 can be seen as invocations of a min $s$-$t$ cut algorithm on $O(\log k)$ many $n$-node graphs; in this claim, we use $\mathcal{A}$ to compute min $s$-$t$ cuts. By union bound, each of those $O(\log k)$ invocations output the desired additive error by probability at least $1 - \alpha O(\log k)$.

