# OpenReview forum: "Nearly Tight Bounds For Differentially Private Multiway Cut"
_NeurIPS.cc/2023/Conference — NeurIPS 2023 oral_

### Official Review · Reviewer_pWxc · 2023-06-23

**Soundness:** 3 good
**Presentation:** 4 excellent
**Contribution:** 3 good
**Rating:** 8
**Confidence:** 3

**Summary:**

The authors give a novel and simple diferentially private algorithm for the min $s$-$t$ cut and multiway $k$-cut problems.

**Strengths:**

Originality:
Work seems original.
Simple but novel!

Quality:
Quality of the paper is high.
The authors give interesting upper and lower bounds for the min $s$-$t$ cut and multiway $k$-cut problems.

Clarity:
Writing is clear.

Significance:
The result is significant, as min $s$-$t$ cut and multiway $k$-cut are important problems.

**Weaknesses:**

None significant weakness found.

**Questions:**

What are some computational complexity implications of your work?

Line 39:
Can you please sketch an example where there are exponentially many min $s$-$t$ cuts but only polynomially many min cuts?

The discussion after Theorem 1.2 (Line 60 to Line 74) is a bit confusing.

Line 65:
Theorem 1.1 should be Theorem 1.2.

Line 89:
Can you elaborate a bit on advanced composition?

Line 161:
Proof of Lemma 3.2:
Can you please add an explanatory figure for the discussion from Line 161 to Line 173?

Line 191:
I do not understand this math display.

Section 4.1 is amazing!

Line 227:
Can you please explain this math display a bit better?

Experiments look OK!

**Limitations:**

Yes.

---

> ### Author Rebuttal · Authors · 2023-08-08
>
> > What are some computational complexity implications of your work?
>
> Thank you for this great question. There are several implications. *First*, our Alg 1 uses a non-private min $s$-$t$ cut algorithm in a black-box manner. Hence, it essentially translates to any popular computation setting without asymptotically increasing complexities compared to non-private algorithms. *Second*, our Alg 2 creates (recursive) graph partitions such that each vertex (and edge) appears in $O(\log k)$ partitions. To see how it differs from other methods, observe that the standard splitting algorithm performs $k-1$ splits, where some vertices appear in each of the split computations. There are also LP-based algorithms for the multiway $k$-cut problem, but to the best of our knowledge, they are less efficient than the $2$-approximate greedy splitting approach. *Third*, in the centralized and parallel settings such as PRAM, in terms of total work, Alg 2 depends only logarithmically on $k$ while the popular greedy splitting algorithm has linear dependence. This question was thoroughly investigated by groups of authors, e.g., see “Faster Parallel Multiterminal Cuts” by Henzinger, Noe, and Schulz, and references therein. To the best of our knowledge, no existing method matches these guarantees that Alg 2 provides.
> We didn’t expand on this in our submission and kept the focus on the DP guarantees of our Alg 2, but we are happy to discuss the complexity implications in more detail in the final version.
>
>
> > Line 39: Can you please sketch an example where there are exponentially many min s-t cuts but only polynomially many min cuts?
>
> Take $n-2$ nodes $v_1,...,v_{n-2}$ in addition to $s$ and $t$, and add $v_i$ to both $s$ and $t$ for all $i$. Suppose the weight of all edges is $1$. There are $2^{n-2}$ min $s$-$t$ cuts, as each min s-t cut should remove exactly one edge from each $s-v_i-t$ path; either $(s, v_i)$ or $(v_i, t)$ is fine.
> In this example, there are $n-2$ min cuts: for each $i$, remove edges $(v_i,s)$ and $(v_i,t)$ and this would be a cut of size 2. Moreover, one can show that the number of min-cuts is polynomial for any graph, there are at most ${n \choose 2}$; please see “On the structure of a family of minimum weighted cuts in a graph” by Dinitz et al.
>
> > The discussion after Theorem 1.2 (Line 60 to Line 74) is a bit confusing.
>
> We will polish this paragraph.
>
> > Line 65: Theorem 1.1 should be Theorem 1.2.
>
> It is true, and we will fix that.
>
> > Line 89: Can you elaborate a bit on advanced composition?
>
> We remark that we do not use the advanced decomposition theorem. We only mention that, even if applied to the standard greedy approach for multiway cut, it does not yield the guarantees that Alg 2 has. To be more precise, the folklore greedy splitting multiway $k$-cut algorithm yields $(k \epsilon, 0)$-DP via direct composition. The advanced decomposition, see [1], improves that to $(\sqrt{k} \epsilon, \delta)$-DP, for some $\delta > 0$. Note that the advanced decomposition yields to approximate DP instead of a pure one in which $\delta = 0$.
> [1] Cynthia Dwork, Guy N. Rothblum, and Salil P. Vadhan. “Boosting and differential privacy”
>
> > Line 161: Proof of Lemma 3.2: Can you please add an explanatory figure for the discussion from Line 161 to Line 173?
>
> Yes, we are happy to improve that discussion.
>
> > Line 191: I do not understand this math display.
>
> Here is a complete derivation
>  $\Pr[w_{G’}(C_u) < w_{G’}(C_v)\ \wedge\ w_{G’}(C_u) < w_{G’}(C_{u,v})\ \wedge\ w_{G’}(C_u) < w_{G’}(C_{\emptyset})] =$
> $\Pr[\Delta_v + X_{s,v} + X_{t,u} < X_{t, v} + X_{s, u}\ \wedge\ \Delta_{u, v} + X_{s, v} + X_{t, u} < X_{t, v} + X_{t, u}\ \wedge\ \Delta_{\emptyset} + X_{s, v} + X_{t, u} < X_{s, v} + X_{s, u}] =$
> $ \Pr[\Delta_v + X_{s,v} + X_{t,u} < X_{t, v} + X_{s, u} \ \wedge\ \Delta_{u, v} + X_{s, v}  < X_{t, v} \ \wedge\ \Delta_{\emptyset} + X_{t, u} < X_{s, u}].$
> Now, we replace $X_{s,v} - X_{t,v}$ by $x$ and $X_{t,u} - X_{s,u}$ by $y$, which leads to the math on Line 191.
> We will include this derivation in the full version.
>
> > Line 227: Can you please explain this math display a bit better?
>
> We provided additional explanations within the Official Rebuttal section. Please check there.

---

### Official Review · Reviewer_PcBW · 2023-07-05

**Soundness:** 3 good
**Presentation:** 4 excellent
**Contribution:** 3 good
**Rating:** 6
**Confidence:** 4

**Summary:**

The paper presents differentially private algorithms for the s-t mincut and multiway cut
problems under an edge privacy model with weights. It shows that a simple algorithm gives
an O(n) additive error bound for the s-t mincut problem, and obtain a near tight lower bound.
This is used for obtaining a bound for the k-multiway cut problem, which has an additional
O(\log{k}) term, which improves over the O(k) term that would result by applying the
s-t cut algorithm repeatedly. The s-t mincut algorithm is evaluated empirically, and the
results are compared with a simple baseline.

**Strengths:**

Differentially private algorithms for the s-t mincut problem have been open for a long time, since
the global mincut algorithm of Gupta et al. The paper gives tight upper and lower bounds on the
complexity of this problem, and shows that a very simple algorithm gives these bounds. For the
k-multiway cut problem, the paper shows that a slightly simple modification of the standard
non-private approximation algorithm can be adapted to have an error with O(log k) dependence,
instead of O(k) dependence. The presentation is good.

**Weaknesses:**

The main ideas in the paper are quite simple. The experiments section is also somewhat limited.
While the presentation is generally good, there is not enough discussion about the related work.
So it is not clear if the technical contributions and the paper overall meet the bar for neurips.

**Questions:**

The edge privacy model assumes the weight of an edge changes by at most 1 in neighboring graphs
(definition 2.3). Suppose this is allowed to be w, do the upper and lower bounds scale accordingly?

Are better accuracy bounds possible when the graphs have some special properties?


line 160: the first sentence ("Let G and G' be two neighboring graphs") doesn't need to be part
of the Lemma statement

It would be interesting to show the relative error of the solution computed by Algorithm 1,
i.e., compare the error of Algorithm 1 with the actual s-t mincut. This could be done for the baseline
as well. Without that it is hard to really understand how well the algorithms are doing.

**Limitations:**

To some extent. There are no negative social impacts

---

> ### Author Rebuttal · Authors · 2023-08-08
>
> > The main ideas in the paper are quite simple. ...
>
> We agree that a separate related work section is beneficial to have all the related references in one place, and we will make one for the final version of our paper. However, there is very limited work on differentially private min-cut and min s-t cut, and we mention most of the references in the introduction. The relevant problems to min s-t cut could be the clustering problems referenced on line 34. However, the techniques for clustering problems are very different from cut problems.
>
> The min-cut problem has a differentially private algorithm that was developed back in 2010 [17] with tight error bound $O(\log(n)/\epsilon)$.
>
> For the multiway-cut problem, we discuss the best approximation algorithms and lower bounds for it in lines 80-83. Several simple algorithms achieve approximation bounds near 2; we mention one of them on lines 85-87. There is also a folklore greedy splitting algorithm (Liang Zhao et al. "Greedy splitting algorithms for approximating multiway partition problems") that starts with the whole graph as one partition and at each step greedily cuts one of the partitions, hence after k-1 rounds achieved k partitions. The problem with all of these algorithms is that they require k rounds of min s-t cut, and hence using an $\epsilon$-private algorithm for min s-t cut in these algorithms would yield a $k\epsilon$-private 2-approximation algorithm for multiway cut. We show that using Algorithm 2, one can decrease this dependency on $k$ to $\log(k)$.
>
> We believe this captures all the related work to our problem. Please let us know if adding any other references or looking into any other problems is beneficial.
>
>
> > The edge privacy model assumes the weight of an edge changes by at most 1 in neighboring graphs (definition 2.3). Suppose this is allowed to be w, do the upper and lower bounds scale accordingly?
>
> This is a great question. Yes, it does generalize to any $w$. Our proof of Lemmas 3.1 and 3.2 is actually already proved for a general weight difference; we use $\tau$ instead of $w$ to define this difference. Those proofs show that our Alg 1 is $(\epsilon \cdot w, 0)$ DP. As a consequence and by rescaling the input $\epsilon$ by $w$, our Alg 1 is $(\epsilon, 0)$ DP and incurs an additive approximation of $O(n \cdot w / \epsilon)$. We are happy to include these comments in the final version.
>
> > Are better accuracy bounds possible when the graphs have some special properties?
>
> This is an interesting question for future research. We are happy to include it in our conclusions. It is worth noting that our lower-bound graph has the min s-t cut equal to 0.
>
> > line 160: the first sentence ("Let G and G' be two neighboring graphs") doesn't need to be part of the Lemma statement
>
> You are right. We will fix this.
>
> > It would be interesting to show the relative error of the solution computed by Algorithm 1, i.e., compare the error of Algorithm 1 with the actual s-t mincut. This could be done for the baseline as well. Without that it is hard to really understand how well the algorithms are doing.
>
> You are right. In the Official rebuttal (please see the attached PDF there) we provided the relative errors. The errors are quite low for both algorithms, though our approach still has lower relative errors.

---

> > ### Comment · Reviewer_PcBW · 2023-08-19
> >
> > Thanks to the authors for the detailed responses to the questions from all the reviewers. I am quite happy with the clarifications

---

### Official Review · Reviewer_u9Zr · 2023-07-06

**Soundness:** 3 good
**Presentation:** 4 excellent
**Contribution:** 3 good
**Rating:** 6
**Confidence:** 3

**Summary:**

This paper presents edge-DP algorithms for min s-t cut and multiway k-cut. The s-t cut algorithm injects fake edges with weights drawn from the exponential distribution, then returns the min s-t cut of the modified graph. The k-cut algorithm is based on $O(\log k)$ calls to the min s-t cut algorithm. They showed that the s-t cut algorithm has $O(n/\varepsilon)$ additive error, along with a lower bound of $n/20$ for $(\varepsilon,\delta)$-DP estimation of the problem, where $\varepsilon \le 1$.

**Strengths:**

- The paper is generally well-structured.
- The DP s-t cut algorithm is surprisingly simple, and it's also the first work on this problem.
- Lower bound is given.


**Weaknesses:**

- The proof of Lemma 4.1 does not match exactly the steps of Algorithm 2. The algorithm could be modified to fit the descriptions in the proof. In any case, I think Algorithm 2 as described, can also be made DP by expending only $O(\log k \cdot \varepsilon)$ privacy, where on each level one can apply parallel composition, similar to what was done in [a].
- There is no related work section? I think the greedy partitioning method for multiway cut [b] could be described here, since it is similar to your non-private multiway cut algorithm.
- The experiments seem uninformative to me. Why compare the error of your s-t cut algorithm to error of a terminal cut? To me, an evaluation of the k-cut algorithm would be more useful, the setup could be: OPT vs. non-private approximation vs. DP algorithm.
- a few minor typos.


[a] Haim Kaplan et al. "Differentially Private Approximate Quantiles"

[b] Liang Zhao et al. "Greedy splitting algorithms for approximating multiway partition problems"


**Questions:**

- The error in Lemma 3.3, I suspect it should be $O(n/\varepsilon \cdot \log n)$? Since the sum of $n$ exponential random variables has gamma distribution with parameters $n$ and $\varepsilon$. I am not sure how you arrived at the bound in line 224, since only an upper bound on the $E[c_i]$ is given.

---

> ### Author Rebuttal · Authors · 2023-08-08
>
> > The proof of Lemma 4.1 does not match exactly the steps of Algorithm 2. ...
>
> We agree with your comment that Lemma 4.1 does not match exactly the steps of Algorithm 2. Our intent is to represent Algorithm 2 in the simplest form, and Lemma 4.1 provides an alternative view on Algorithm 2, which we then leverage to prove DP. We agree with your comment on how DP guarantees can be proved, and our formal proof – please see Theorem 4.2 – is very similar to the argument you provided.
>
> > There is no related work section? I think the greedy partitioning method for multiway cut [b] could be described here, since it is similar to your non-private multiway cut algorithm.
>
> Thank you for providing the references, we will mention the references you suggested. The greedy splitting technique is indeed one of the most widely known algorithms. However, there is a _significant_ difference between the greedy splitting and our approach.
> Namely, the usual greedy splitting has $k$-dependant splits, while ours has only $O(\log k)$. That $k$ dependence also applies to algorithms GSA and M-GSA in the reference [b]. As a result, we get much stronger DP guarantees compared to the direct application of the folklore greedy splitting yields. As an additional feature, the running time of our algorithm depends only logarithmically in $k$, as opposed to linearly like the algorithms in [b].
>
> Even though Algorithm 2 is simple and intuitive (and similar results have been proved for other problems such as in [a]), our proof is more technically involved and quite different from that of [b].
> The same problem described above exists for the 2-approximation algorithm outlined in lines 85-87 of our submission.
>
> We agree that a separate related work section is beneficial to have all the related references in one place and we will make one for the final version of our paper. However, there is very limited work on differentially private min-cut and min s-t cut, and we mention most of the references in the introduction. In addition to the references provided, please let us know if adding any other references or looking into any other problems is beneficial.
>
> > The experiments seem uninformative to me. Why compare the error of your s-t cut algorithm to error of a terminal cut? To me, an evaluation of the k-cut algorithm would be more useful, the setup could be: OPT vs. non-private approximation vs. DP algorithm.
>
> The terminal cut is the only differentially private benchmark we are aware of. Another potential benchmark is partitioning the vertices randomly without even considering edges. However, although it has perfect privacy, such a benchmark has an extremely high additive error of the order $0.5$ * # of edges. We provide the relative errors in the PDF in the Official Rebuttal, showing that the relative errors are very small.
>
> For $k>2$, obtaining the optimum solution is NP-hard, so experimenting with that is more tricky. Moreover, we focused on experimentally evaluating Alg 1 because our Alg 2 achieves DP by using Alg 1 as a sub-routine.
>
> > The error in Lemma 3.3, I suspect it should be $O(n / \epsilon \cdot \log n)$? ...
>
> We still believe the right answer is $O(n / \epsilon)$. Indeed, it is more involved to show that bound than $O(n \log n / \epsilon)$; the bound of $O(n \log n / \epsilon)$ is direct as each of the $n$ random variables has value $O(\log n / \epsilon)$ with high probability. Intuitively, for a sufficiently large constant $c$, one can still show a bound of $O(n / \epsilon)$ because only _a small fraction_ of random variables exceeds $c/\epsilon$; this fraction is roughly $e^{-c}$. We explicitly state this on Line 220 and in Fact 2.1, and then leverage it in our proofs. In the Official rebuttal for the entire submission, we comment more on that paragraph, including expanding on Line 224. We are happy to provide more details in that proof in the final version.

---

> > ### Comment · Reviewer_u9Zr · 2023-08-16
> >
> > Thank you for your response.
> >
> > You're right, obtaining the optimal solution for general $k$ is NP-hard, but using an approximation algorithm for the experiments would also suffice. Because the main theme is "multiway" cut, and one of the stated contributions is in reducing the error from $k$ to $\log k$. I think experimental results which show how the error grows with $k$ would be better support for this claim.
> >
> > I will keep the score.

---

### Author Rebuttal · Authors · 2023-08-05

We thank all the reviewers for their constructive comments.

Two of the reviewers asked for clarification of the analysis in Lemma 3.3, so in addition to our comments below, we provide more explanations that we will include in the final version.

For intuition and brevity, assume that $\epsilon = 1$. Observe that showing that Alg 1 has $O(n \log n)$ additive error is straightforward as a random variable drawn from $Exp(b)$ is upper-bounded by $O(\log n / b)$ whp. Our Lemma 3.3 proves the $O(n)$ bound. It is instructive to think of the interval $[2, 5 \log n]$ being partitioned into buckets of the form $[2^i, 2^{i+1})$. Then, the value of each edge added by Alg 1 falls into one of the buckets; we can safely assume that no edge $X_{s, u}$ or $X_{t, u}$ has a weight more than $5 \log n$. Now the task becomes upper-bounding the number $c_i$ of edges in bucket $i$. That is, we let $Y_{s, u}^i = 1$ iff $X_{s, u} \in [2^i, 2^{i+1})$, which results in $c_i = \sum_{u \in V}(Y_{s, u}^i + Y_{t, u}^i)$. Hence, $c_i$ is a sum of 0/1 independent random variables, and we can use Chernoff bound to argue about its concentration.
There are two cases. If $E[c_i]$ is more than $O(\log n)$, then $c_i \in O(E[c_i])$ by the Chernoff bound. If $E[c_i] \in o(\log n)$, e.g., $E[c_i] = O(1)$, we can not say that with high probability $c_i \in O(E[c_i])$. Nevertheless, we can still say $c_i \in O(\log n)$ whp. This is why there is $\max$ including $\log n$ in it.
Finally, Line 227 simply sums the weights of $X$ random variables over all the buckets, $\log (5 \log n)$ of them, where if a random variable $X$ falls into bucket $i$ we upper-bound its weight by $2^{i + 1}$, although it can be smaller by a factor of $2$. Also, to obtain the first inequality on Line 227, we use that $\max(\log n, 2n / 2^{2i}) \le \log n + 2n / 2^{2i}$.

In regards to the experiments, we attached a figure containing relative errors as requested by some of the reviewers. This figure shows that the relative errors are very small. Note that the errors are computed with respect to the optimal $s$-$t$-cut which is not private.

---

### Decision · Program_Chairs · 2023-09-21

**Decision:**

Accept (oral)

**Comment:**

The paper presents new differentially private algorithms for s-t mincut and multiway cut problems. The problem is interesting and the techniques are novel. The reviewers were in agreement that the paper is above bar and I am happy to recommend acceptance. The authors are encouraged to incorporate reviewer feedback, and relevant portions of the rebuttal in the final version of the paper. It would be great also if they could further elaborate on the relevance of their solutions to various ML problems.